# Statistical learning goes beyond the *d*-band model providing the thermochemistry of adsorbates on transition metals

Rodrigo García-Muelas [1]* & Núria López [1]*

The rational design of heterogeneous catalysts relies on the efficient survey of mechanisms by density functional theory (DFT). However, massive reaction networks cannot be sampled effectively as they grow exponentially with the size of reactants. Here we present a statistical principal component analysis and regression applied to the DFT thermochemical data of 71 $C_1$–$C_2$ species on 12 close-packed metal surfaces. Adsorption is controlled by covalent (*d*-band center) and ionic terms (reduction potential), modulated by conjugation and conformational contributions. All formation energies can be reproduced from only three key intermediates (predictors) calculated with DFT. The results agree with accurate experimental measurements having error bars comparable to those of DFT. The procedure can be extended to single-atom and near-surface alloys reducing the number of explicit DFT calculation needed by a factor of 20, thus paving the way for a rapid and accurate survey of whole reaction networks on multimetallic surfaces.

[1] Institute of Chemical Research of Catalonia (ICIQ), The Barcelona Institute of Science and Technology (BIST), Av. Països Catalans 16, 43007 Tarragona, Spain. *email: rgarcia@iciq.es; nlopez@iciq.es

Heterogeneous catalysis holds the key to solving fundamental sustainability issues by introducing renewable compounds as a source of chemicals and energy vectors[1–3]. The rational search for new catalysts benefits from the extensive use of density functional theory (DFT) and kinetic models derived from it[3–10]. This procedure requires sampling the reaction network that links reactants, intermediates, and products through transition states. For large molecules, such as those involved in biomass valorization processes, the number of intermediates and transition states grow exponentially with the molecular size, rendering the computational screening of new catalysts impractical. For instance, the decomposition of a medium size $C_5$–$C_6$ sugar alcohols encompasses more than $10^4$ species and $10^5$ transition states in its reaction network[11]. The positive aspect is that kinetic barriers are linked to the formation energies of different intermediates[4,12]. Thus, the screening may be reduced to obtain the thermodynamics for the adsorbed species and couple them with linear-scaling relationships and microkinetic models to simulate operation conditions. The upgrade of biomass-derived molecules is often done by metals and alloys[1,13], and lately attention has been drawn to the versatile properties of single-atom alloys (SAAs) and near-surface alloys (NSAs)[9,14–22]. The number of combinations is again unlimited and some have shown an almost continuum of adsorption strengths[21]. To this end, new thermochemical models based on statistical learning may allow a rapid survey of the energies of adsorbed species for faster screening[8,18,23–27].

The pioneering work by Benson established the basis for thermochemical scaling relationships of gas-phase molecules already in the 60s[28]. In this formulation, the formation energy for a hydrocarbon or oxygenated molecule is obtained as the sum of the energies stored on C–C, C–O, C–H, and O–H bonds, considering also the contribution from rings, unsaturations, and radicals[28]. Despite its simplicity, Benson's model has an impressive accuracy for small molecules such as hydrocarbons, alcohols, and ethers, the formation energy of which is predicted with errors lower than 0.05 eV[29].

When molecules adsorb on metal surfaces, the interaction has covalent, ionic, and dispersion contributions. The most studied term is the covalency appearing from the coupling of the metal $sp$- and $d$-states with the adsorbate. The $sp$ part depends on the species but it is rather constant along the metals. The second one gives the metal-to-metal variability and comes from the $d$-band center and filling[30,31]. As a consequence, the adsorption energy of a molecular fragment $AH_x$ is a linear function of that of its heteroatom, A, and the slope accounts for the valence of $AH_x$[32]. Further linear dependencies have been identified for heteroatoms belonging to the same group in the periodic table, i.e., P* scales with N*[33]. In addition, the accuracy of the above models can be improved by using site-specific adsorption rules[22,32] and the dependence on the local coordination of the adsorption sites[22,34,35]. In the particular case of very small nanoparticles, the activity modulation is linearly dependent with the local electrostatic potential[36].

These adsorption energy models have been extended to multifunctionalized molecules by combining the heteroatom scalings and the Benson model[28,37,38]. The combination can be centered either on individual bond energies[37] or on the coordination environment of each heteroatom[38]. Attempts to generalize simplified thermochemical models to other materials are less frequent, but for perovskites and transition metal oxides[39–41], electronic parameters such as the occupancy of $e_g$ orbitals and the covalency for the oxygen-transition metal bond were deemed descriptors for their catalytic activity.

Still, the reactivity on metals has provided the largest amount of DFT data and benchmarks on the thermochemistry

demonstrate the robustness of the results[42]. Thus, large FAIR databases[43] open alternative paths to rationally design heterogeneous catalysts[10], by improving existing thermochemical models and generating new ones through statistical learning. For instance, the formation energies of large molecules in gas phase can be retrieved from neural networks with an accuracy comparable to DFT, 0.04 eV MAE[44], thus beyond Benson's model[29]. An alternative approach is to predict the formation energy of few reactivity descriptors from geometric and electronic features, such as atomic radius, local electronegativity, ionic potential, and the coordination of the active site[9,18,22,45]. Particularly, a recent SISSO study considers features from the adsorbate, the metal, and the adsorption site[45], providing excellent predictions with one DFT evaluation for each metal. However, the method does not belong to the explanatory class of machine-learning techniques and thus results are difficult to interpret. A key step to generalize statistical learning models is to extract physical insights from them. For instance, a feature importance analysis[18] rediscovered the differentiated roles of $d$- and $sp$-band contributions[31,46]. Other studies have highlighted the role of electronic and redox descriptors for the thermochemistry of transition metal complexes[47] and oxide-supported single-metal atoms[25]. Yet, the potential of statistical learning in heterogeneous catalysis remains largely unexplored[9,26].

In the present work, we applied principal component analysis and Regression (PCA, PCR) on a set of formation energies obtained by DFT. From the descriptors so obtained, we retrieved the $d$-band center and the redox ability of the metal as the main controllers of the thermochemistry, along conjugation and conformational effects. With these descriptors and a minimum set of DFT energy evaluations (around two-thousand), we predicted a full thermochemical database of 31,000 species adsorbed on pure metals, SAAs and NSAs. The methodology reduces the number of explicit DFT evaluations by a factor of 20 keeping its accuracy. As the procedure is modular it can be adapted or extended to other systems in heterogeneous catalysis.

## Results

**Interpretation of thermochemical data by PCA.** The first step consists in the generation of a well-converged database of formation energies on late transition metals: Cu, Ag, Au, Ni, Pd, Pt, Rh, Ir, Ru, Os, Zn, and Cd. This was done using the PBE-D2 functional following the gold standard in DFT[42]. The formation energies, $E_{C_xH_yO_z*}$, are referred to gas-phase reservoirs of methane, hydrogen, and water, Eqs. (1)–(2). In all cases, the lowest energy conformation was employed to ensure that the PCA includes the information corresponding to conjugation and conformational changes. The data are arranged in a matrix-**E** in which the rows span the metals $i$ and the columns correspond to the adsorbates $j$. As the data matrix is complete, meaning that it does not have any missing points, it is suitable for PCA. PCA is a statistical technique that reduces the dimensionality of **E** by projecting it along the directions of greatest variability, thus reducing its noise and aiding to its interpretability[48]. The process, summarized in Fig. 1 and detailed in the Supplementary Methods, proceeds as follows: the average adsorption energy for each intermediate, $\mu_j$, is employed to center the adsorption matrix-**E** and get **X**. We kept the units of **X** as eV. This matrix is multiplied at the left by its own transpose to get the covariance matrix **C**, which is then diagonalized. The eigenvalues of the diagonal matrix **D** are all positive and are placed in decreasing order, consistently with the eigenvector matrix **V**. Afterwards, **V** is truncated to $k_{max}$ principal components and multiplied at the left by **X** to get **W** and **T**. These matrices contain the descriptors for the metals and adsorbates, $t_{ik}$ and $w_{kj}$, respectively. The

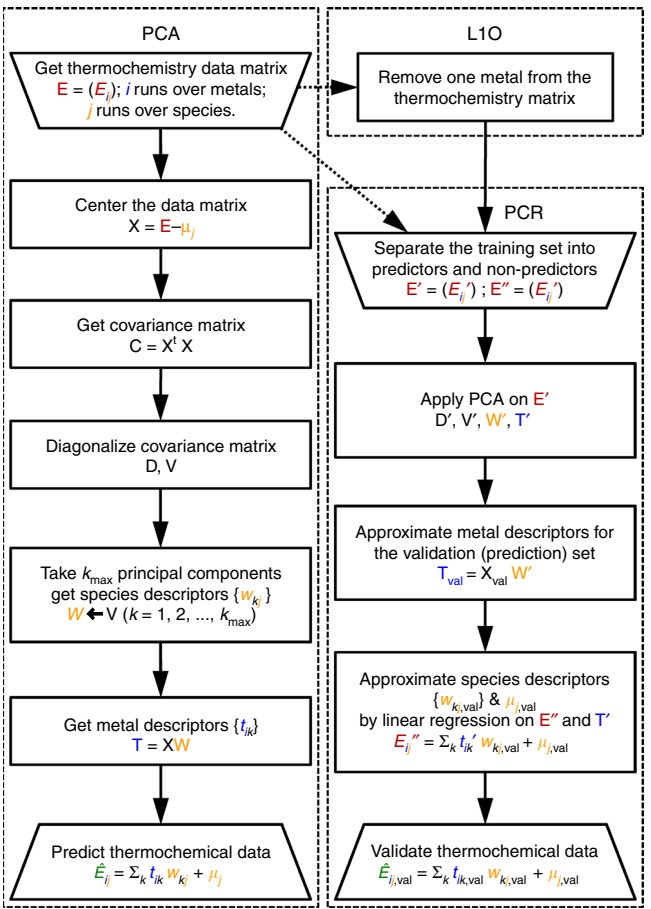

**Fig. 1** Principal component analysis (PCA) and regression (PCR). The formation energy of species $j$ on metal $i$ is obtained from DFT and Eqs. (1)-(2). These energies are grouped in thermochemistry matrices, in red, and are approximated following PCA or PCR, in green. PCR was validated by leaving-one-metal-out of the data matrix (L1O). Variables associated with metals and species are shown in blue and orange, respectively. In black, variables associated with mathematical procedures. A data flow diagram, including the sizes of all matrices in this study, is shown in Supplementary Fig. 3

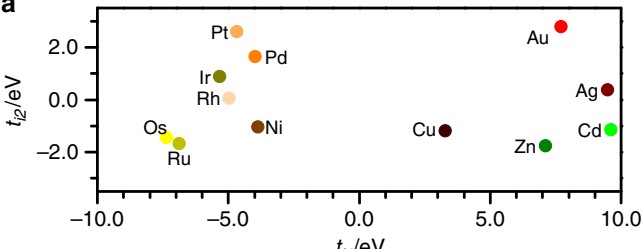

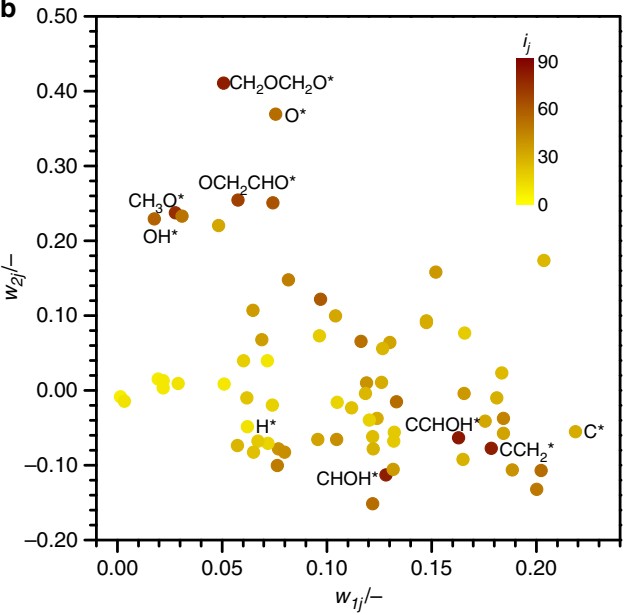

**Fig. 2** Descriptors from principal component analysis. Descriptors for **a** metals $\{(t_{i1}, t_{i2})\}$ and **b** adsorbates $\{(w_{1j}, w_{2j})\}$, obtained by PCA. The color scale in **b** measures the robustness of each species of being a predictor ($\iota_j$). Those marked in brown are more suitable predictors, as they have large projections in at least one $w_{ik}$ and a low SD, Supplementary Eq. 12. Those marked in yellow are the least suitable. Relevant species are labeled

adsorption energies can then be retrieved from Eq. (3).

$$x\text{CH}_4 + \left(-2x + \frac{1}{2}y - z\right)\text{H}_2 + z\text{H}_2\text{O} +^* \longrightarrow \text{C}_x\text{H}_y\text{O}_z^* \quad (1)$$

$$E_{\text{C}_x\text{H}_y\text{O}_z^*} = E_{\text{C}_x\text{H}_y\text{O}_z^*}^{\text{vasp}} - xE_{\text{CH}_4}^{\text{vasp}} + \left(2x - \frac{1}{2}y + z\right)E_{\text{H}_2}^{\text{vasp}} - zE_{\text{H}_2\text{O}}^{\text{vasp}} - E_*^{\text{vasp}} \quad (2)$$

$$\hat{E}_{ij} = t_{i1}w_{1j} + t_{i2}w_{2j} + \cdots + t_{ik}w_{kj} + \cdots + t_{ik_{max}}w_{k_{max}j} + \mu_j \quad (3)$$

The accuracy of Eq. (3) is given by the number of principal components selected; this is, the number of $t_{ik}w_{kj}$ terms. To assess the minimum number of terms in the expansion, $k_{max}$, we have evaluated two criteria: the MAE and the variance, see Supplementary Table 3. In the first case, the MAE stagnates for two terms to a value that falls within the DFT accuracy. At that point, 98.1% of the variance is already captured. As a result, only two principal components, $k_{max} = 2$, will be used from now on and only two descriptors are needed for metals $\{(t_{i1}, t_{i2})\}$ and adsorbates $\{(w_{1j}, w_{2j})\}$, Fig. 2a, b. These descriptors wrap up the causes of variability in metal-adsorbate bond energies.

The first metal descriptor, $t_{i1}$ spans the larger variability, 17 eV, whereas the second one, $t_{i2}$, accounts only for one-third of this energy span, Fig. 2a. The metals tend to be ordered left-to-right by their position on each period in the table of elements. Those more resistant to oxidation, such as Pt and Au, appear on the topmost region.

All adsorbates appear in the right side of Fig. 2b, meaning that their first weight, $w_{1j}$, is always positive. The largest terms belong to highly unsaturated carbonaceous species. Therefore, the first term, $t_{i1}w_{1j}$, can be interpreted as the affinity of metal $i$ to form covalent bonds with intermediate $j$. For late transition metals (groups 8–11), this characteristic can be mapped to the $d$-band center, Fig. 3a, which modulates the adsorption strength on such surfaces[30,31]. However, the $d$-band model cannot describe several systems that are relevant for catalysis: it does not apply for adsorbates with an almost completely filled valence shell, such as *OH, adsorbing on alloys with almost-filled $d$-bands[46]. Besides, the $d$-band model cannot describe metals from group 12 (Zn, Cd) and above, which can be components in high-entropy alloys suited for electrocatalysis[21]. The second weight for the adsorbates, $w_{2j}$, is positive for those that bind through an oxygen atom, such as O* and *OCH$_2$CH$_2$O*, and negative in species that bind by a *COH center. Therefore, the second term, $t_{i2}w_{2j}$, can be interpreted as the ionicity of the metal-adsorbate bond. As such,

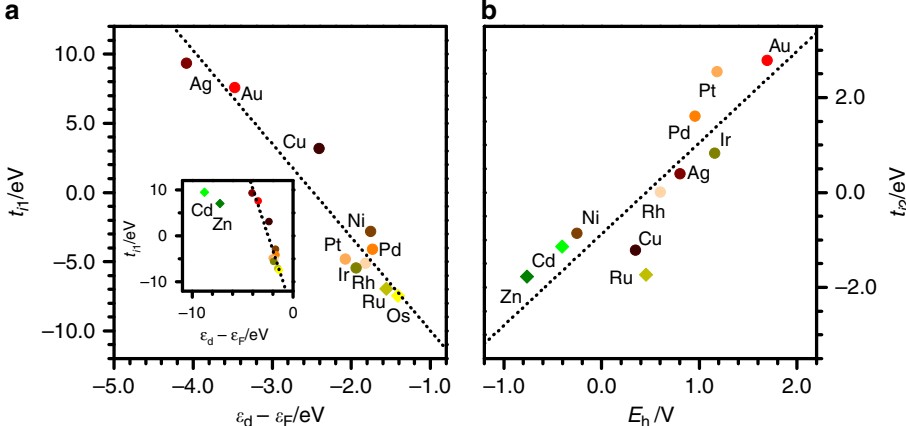

**Fig. 3** Interpretation of metal descriptors in physical terms. First and second descriptors for the metals, $t_{i1}$ and $t_{i2}$, plotted against **a** the *d*-band center and **b** the reduction potential[50], respectively. The inset shows that *d*-band center controls the adsorption energy on late transition metals (groups 8–11)[30,31], but it cannot describe the behavior of Zn and Cd. Additional data and plots are provided in Supplementary Table 2 and Supplementary Figs. 1 and 5

$t_{i2}$ can be mapped to the reduction potential, Fig. 3b. Cross-correlations do not appear, Supplementary Fig. 1, meaning that $t_{i1}w_{1j}$ and $t_{i2}w_{2j}$ summarize rather independently the covalent and ionic contributions, respectively.

The fact that 98.1% of the variance is captured by the aforementioned descriptors, implies that other contributions to the thermochemistry, such as conjugation, conformational changes (different adsorption sites), and dispersion (van der Waals) are already included, as they are related to the two major covalent and ionic terms. For instance, $CHCH_2$ can adsorb as monodentate ($*CH=CH_2$), tridentate ($**CH–*CH_2$), or intermediate structures. The most stable conformation depends on the metal affinity to carbon, Supplementary Fig. 4. This means that the adsorbate valence is not necessarily an integer, as it is normally assumed in heteroatom scaling relationships[32]. Also, small molecules such as $*OH$ can adsorb on fcc, hcp, bridge, and top-tilted sites[49], and the preferred site can differ even for chemically similar metals, such as Rh (fcc sites) and Ir (bridge), or Ru (hpc) and Os (bridge). In other words, the most stable conformation of the adsorbate is defined by the metal, thus highlighting the interplay of the metal-adsorbate system.

To provide a rapid survey on the adsorption energies of surface species, it is desirable to calculate by DFT only a small subset of intermediates, called predictors. Their number should be at least equal to the number of principal components $k_{max}$[27,48]. Choosing the predictors is not evident from Fig. 2b alone. For instance, the simplest set would contain only the heteroatoms $C^*$ and $O^*$[27], but $E_O$ and $E_C$ are mildly codependent. This codependence appears in many pairs of adsorbates, as can be seen in Supplementary Fig. 2, although its origin was unknown[33,51]. Therefore, the predictor set can be expanded to ensure that the full DFT database, (**E** in Fig. 1), is properly estimated ($\hat{\mathbf{E}}$).

To select the proper predictor set, we calculated the error matrix ($\hat{\mathbf{E}} - \mathbf{E}$) that compares the potential energies obtained by DFT to the ones estimated by Eq. (3). Then, the robustness of each intermediate as a predictor, $\iota_j$, was measured following the procedure detailed in Supplementary Methods. This $\iota_j$ is indicated by the color scale in Fig. 2b, in which dark brown indicates at least one large $w_{kj}$ and a low SD. Therefore, such species are better at predicting the thermochemistry of others. Interestingly, we noticed that when the prediction error of $O^*$ was positive, the one of $*OH$ tends to be negative and vice versa. We also chose $*CCHOH$ as predictor, as it has a pair ($w_{1j}, w_{2j}$) that is almost orthogonal to $O^*$ and $*OH$, Fig. 2b. In summary, the full

thermochemistry of a given metal can be estimated from two principal components obtained from the formation energies of three[27] predictors ($O^*$, $*OH$, and $*CCHOH$) that capture most of the variability of the original data matrix. The principal components define two descriptors for both metals ($t_{i1}$, $t_{i2}$) and molecules ($w_{1j}$, $w_{2j}$).

**Fast and accurate prediction of thermochemistry via PCR.** To assess the accuracy of the statistical learning tools, we performed two set of tests, using PCA and PCR-L1O, respectively (Fig. 4). In both cases, we compared the results of the estimated formation energies from Eq. (3) ($\hat{\mathbf{E}}$, Fig. 1) with those obtained by DFT (**E**, Fig. 1). For PCA, the training and prediction sets are equal, as they contain the full DFT set: 71 molecules and 12 metals. PCA estimates all the energies within ±0.50 eV and 98% of them lie within ±0.30 eV, with a MAE of 0.08 eV (Fig. 4a, b). The test on the PCR is stricter as the training set is reduced to only three molecules as predictors. We followed a leave-one-out (L1O) validation in which we took a subset of 11 metals (training set) to predict the thermochemistry of the 12th one (validation set) (Fig. 1). This matrix of energies is split into two submatrices containing just the three predictors (**E′**) and the remaining species (**E″**). In total, 792 DFT evaluations are required to get **E′**, corresponding to 11 metals × 71 species plus the empty surface. For the validation set (**E**$_{val}$), only four DFT evaluations are needed and correspond to the clean surface and the three predictors. The PCR-L1O starts by applying PCA on the **E′** submatrix to obtain **T′** and **W′**. Then the descriptors for the metals in the validation set, $t_{1,val}$ and $t_{2,val}$, are estimated from the DFT formation energies of the three predictors. The descriptors for the remaining species ($w_{1j,val}$, $w_{2j,val}$, and $\mu_{j,val}$) are then found via linear regression of Eq. (4) on the training set. Finally, the thermochemistry of the validation set is predicted from Eq. (5). This procedure is sequentially run to consider every metal independently as a validation set. Fig. 4c, d compares the results from PCR-L1O with DFT data, showing that the MAE increases to 0.12 eV and the population in the central bars is about 25% smaller than for PCA. Still, 98% of the estimated energies lie within ±0.40 eV. Thus, the PCR-L1O methodology only increases the error span by 0.10 eV. If $C^*$ and $O^*$ were used as predictors instead of $O^*$, $*OH$, and $*CCHOH$, the MAE would have rise to 0.16 eV and the maximum error to ±1.00 eV. Writing the formation energies as linear regression of the *d*-band center and the oxidation potential (excluding Os, Cd, and Zn) would increase the MAE to 0.18 eV

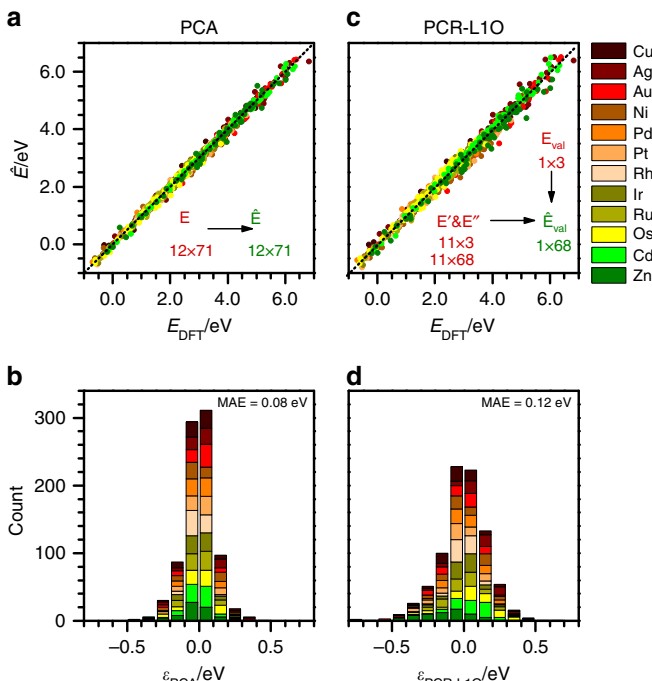

**Fig. 4** Accuracy assessment of PCA and PCR. **a** Error distribution and **b** cumulative errors for the PCA, taking the DFT data of the 71 molecules on the 12 metals. **c**, **d** The corresponding values for PCR-L1O, using only O*, *OH, and *CCHOH as predictors, and leaving-one-metal-out of the pool (L1O). The inset shows the size of the corresponding input (red) and output (green) energy matrices

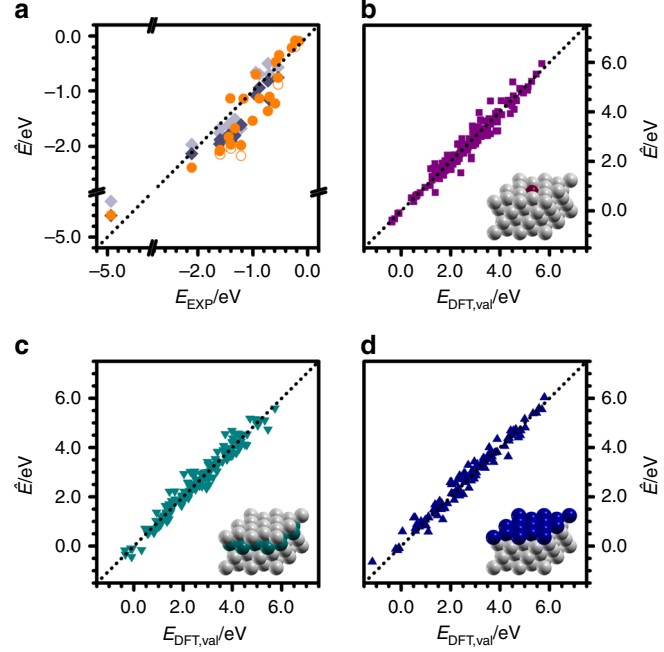

**Fig. 5** Prediction of adsorption energies on metal alloys through PCR. **a** PCR-L1O estimates of formation energies plotted against experimental data of Supplementary Table 4 (orange circles)[53]. Open circles stand for the predicted values when coverage effects are not considered. DFT data from ref. [52] are provided as benchmark for PBE and BEEF-vdW density functionals, in dark and light gray, respectively. **b** Single-atom (SAA), **c** sub-surface (NSA-SS), and **d** overlayer (NSA-OL) alloys. Taking the DFT thermochemistry of 71 adsorbates on 12 pure metals and 3 predictors for each alloy (*O, *OH, and *CCHOH), the adsorption energies for the remaining 68 adsorbates was predicted. Then, 1% of these species were randomly selected as validation set, calculated by DFT, and benchmarked on **b**–**d**

and the maximum error to ±0.80 eV. Thus, higher accuracies can be obtained from predictors calculated by DFT than by using tabulated data.

$$E''_{ij} = t'_{i1} w_{1j,\mathrm{val}} + t'_{i2} w_{2j,\mathrm{val}} + \cdots + t'_{ik_{\max}} w_{k_{\max}j,\mathrm{val}} + \mu_{j,\mathrm{val}} \quad (4)$$

$$\hat{E}_{ij,\mathrm{val}} = t_{i1,\mathrm{val}} w_{1j,\mathrm{val}} + t_{i2,\mathrm{val}} w_{2j,\mathrm{val}} + \cdots + t_{ik_{\max},\mathrm{val}} w_{k_{\max}j,\mathrm{val}} + \mu_{j,\mathrm{val}} \quad (5)$$

The adsorption energies of metals that are better predicted correspond to Rh and Ir with 0.05 and 0.08 eV MAE, whereas the worst are Zn and Ag with 0.18 and 0.17 eV MAE. Zn (Ag) has the more exothermic (endothermic) average of formation energies. Therefore, the binding energy of their adsorbates is normally underestimated (overestimated). In general, the predicted values are least accurate for species that have highly endothermic formation energies, such as *CHOHCHOH and *CCH. However, they would play a minor role when large reaction networks are taken as a whole particularly when different competitive paths are wrapped through microkinetic models[7]. The PCR-L1O method was then benchmarked against experimental and DFT thermochemical data (Fig. 5a)[52,53]. Our estimates, shown in orange, lie nicely close to the 1:1 line. The error bars lie between the ±0.25 eV typical of DFT predictions, as shown by PBE and BEEF-vdW values obtained by other groups (in dark and light gray, respectively)[52,53].

Finally, we have used PCR to generate the first full thermochemical database containing SAAs and NSAs. The SAAs were obtained by replacing one atom by the guest element, whereas the NSA[14] were generated by substituting either the overlayer or the sub-surface layer (Fig. 5b–d). The host elements were those listed in Fig. 4 and the guest elements also included Fe, Co, and Re, for a

total of $12 * (15 - 1) = 168$ SAA and 336 NSA. For each alloy, the prediction of the thermochemistry required four DFT evaluations, corresponding to the clean surface and three predictors: O*, *OH, and *CCHOH. The alloys whose structures did not converge are listed in the Supplementary Methods and were removed from the pool, leaving 165 SAAs and 278 NSAs. These results, collected on matrix $\mathbf{E}_{\mathrm{val}}$, required 1772 DFT evaluations. With this data, and taking the pure metals as a training set ($12 \times 71 = 852$ DFT evaluations distributed between matrices $\mathbf{E}$ and $\mathbf{E}'$), a PCR was applied to predict the thermochemistry for the remaining 68 species on each alloy. As a result, a database of 31,453 formation energies was generated (Supplementary File: alloys-prediction.csv). To benchmark this database, 1% of these species were randomly selected, calculated by DFT, and compared with the predictions of PCR. The benchmark, shown in Fig. 5b–d excluded those species that belong to predictor set. In all cases, the estimates are within the DFT accuracy, with a MAE around 0.19 eV. This shows the high predictive power of PCR as the number of explicit DFT calculations is reduced 20 times for each new alloy. However, the PCA/PCR employed above can be extended to account for other effects, likely appearing as new components in the expansion in Eq. (5). Examples of these effects are coverage contributions[54], transition state search[18], site coordination[22,35,45], and solvation[7,55].

## Discussion

Statistical learning provides a robust toolbox to go beyond the traditional interpretation of the energetics of intermediates on transition metal surfaces. By applying PCA on a set of formation

energies obtained by DFT, we extracted only two descriptors for the reactivity of metals and alloys. A close analysis to these descriptors revealed their nature as covalent and ionic contributions, which can be traced back to the well-known $d$-band model, the reduction potential of the metal, as well as conjugation and conformational effects. Then, PCR enabled us to do a fast survey on the adsorption on SAAs and NSAs, using just three adsorbates as predictors: *O, *OH, and *CCHOH. A minimum of DFT energy evaluations (around 1800) is required to predict the full set of 31,000 formation energies with high accuracy, as the error bars are comparable to DFT ones. This approach is modular and can be extended to other materials and systems, such as the prediction of activation energies for elementary steps or the quantification of solvent and coverage effects, thus paving the way for reliable thermochemical models suited to heterogeneous catalysis.

## Methods

**Computational details**. We performed DFT calculations with the Vienna Ab-initio Simulation Package, VASP[56,57], for 71 species derived from $C_1$ and $C_2$ alcohol decomposition[7,49] on closed-packed surfaces of Cu, Ag, Au, Ni, Pd, Pt, Rh, Ir, Ru, Os, Zn, and Cd. The structures can be retrieved from ioChem-BD[58,59], which is a FAIR (Findability, Accessibility, Interoperability, and Reusability) database[43]. Instructions about data management are provided in Supplementary Methods. The functional of choice was PBE[60] with the D2 dispersion corrections of Grimme and our reparameterized values for metals[61,62]. The structural and electronic parameters for the metals can be found on Supplementary Tables 1–2. The present setup follows the current gold standard in DFT calculations[42]. Core electrons were represented by Projector Augmented Wave pseudopotentials[63,64] and valence electrons were represented by plane waves with a kinetic energy cutoff of 450 eV. The calculated lattice parameters for the metals show good agreement with experimental values, as detailed in the Supplementary Information. Metal surfaces were modeled by four-layer slabs, where the two uppermost layers were fully relaxed and the bottom ones were fixed to the bulk distances. We selected the (111) surfaces for the fcc metals and the (0001) for the hcp ones. The adsorption was studied on $2\sqrt{3} \times 2\sqrt{3} - R30°$ supercells. The vacuum between the slabs was set larger than 13 Å and the dipole correction was applied in $z$ direction[65]. The Brillouin zone was sampled by a $\Gamma$-centered $3 \times 3 \times 1$ $k$-points mesh generated through the Monkhorst–Pack method[66]. For each species, several conformations were calculated by DFT[7,49], but only the most stable ones were taken for subsequent analysis. The gas-phase molecules were relaxed in a cubic box with 20 Å sides. For PCA and PCR, the diagonalizations were done with Maple using double precision.

## Data availability

All relevant data are available from the authors. The matrices **E**, **E'**, and **E''** from the training set are uploaded as Supplementary Files matrix-E.csv, matrix-E-prime.csv, and matrix-E-second.csv, respectively. The matrix **E'** for SAAs and NSAs is presented on matrix-E-prime-alloys.csv and the 31,453 structures predicted are listed on alloys-prediction.csv. The matrix files are labeled using a succinct notation detailed in Supplementary Methods and Supplementary Table 5. The structures of all species can be downloaded from ioChem-BD[58] following ref. [59].

## Code availability

The LibreOffice Calc spreadsheets, Maple worksheets, and Python scripts are available from the authors. The data from pure metals and alloys are processed in pca-regressions.ods and pca-alloys.ods spreadsheets, respectively. The Maple worksheet used for diagonalizations is diagonalization.mw. The python script that generates all thermochemical data for alloys is pca-alloys.py.

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

## Acknowledgements

We thank AGAUR 2017 SGR 90 and MCIU RTI2018-101394-B-I00 projects for financial support. The Barcelona Supercomputing Center – MareNostrum (BSC-RES) is acknowledged for providing generous computer resources. We thank Professors C. Bo, R. Guimerà, M. Sales-Pardo, F. Abild-Pedersen, and N. Almora-Barrios for fruitful discussions.

## Author contributions

R.G.-M. performed the numerical calculations. R.G.-M. and N.L. contributed to analyze the data and to write the manuscript.

## Competing interests

The authors declare no competing interests.
