## [Peer Review File · Nature Communications]

Reviewers' comments:

Reviewer #1 (Remarks to the Author):

A new thermochemical model based on statistical learning is developed for rapid survey of adsorption energies of complex chemistry on metal surfaces. The d-band center and the redox ability were retrieved as main controllers of the thermochemistry. With those two descriptors along with 2000 DFT evaluations, a full database of 31,000 species adsorbed on metals, single-atom and near surface alloys were predicted.

The work is an extension of scaling relations and group additivity concepts popularized in heterogeneous catalysis using machine learning. The model prediction compared to just physically selected groups with energy corrections due to conformation and conjugation.

The chemistry involved in upgrading biomass-derived molecules is complex. The approach is not demonstrated to be useful for catalyst design rather than showing the prediction of binding energies of big fragments using simple groups, e.g., *C, *OH, and *OCH₂CH₂O*.

PCA analysis is not neat for dimensionality reduction. The two principle components that maximize the variance of adsorption energies are correlated to the d-band center and reduction potentials, respectively. The result is very interesting. However, relating the two descriptors of surfaces to two characteristics of metals is arbitrary and not necessary captures underlying physics.

On page 8, it states that the conjugation and conformational effects are already wrapped up in the two descriptors. Please explain clearly.

Three components are chosen later for capturing a broad range of systems. But on page 9, it says the estimation can be done with two principle components obtained from the formation energies of three surface species. It is not clear what they mean. To me, the number of descriptors for surface has to be same length as the number of surface species required for estimation.

Reviewer #2 (Remarks to the Author):

The authors present an approach for estimating adsorption of organics onto metals, alloys and single atoms based upon a principle component analysis/regression of DFT binding energies. Their analysis points to the need for only a few descriptors (d-band center, work function) to accurately (+/-0.3eV over a span of <10eV) reproduce full DFT results. Given all the work over the years on scaling relationships it was obvious that something like this was at play, but these are the first authors to formalize it in a convincing way. I think a better description on the alloys and single atoms would help understand what they did-the presentation is vague to the point where we do not know what systems etc. Overall the paper is fine and on a topic that is clearly in vogue.

Reviewer #3 (Remarks to the Author):

In this work, the authors employ DFT and principle component analysis/regression (i.e., machine learning, ML) to extract data-driven descriptors for predicting C1-C2 adsorption trends on the surfaces of both pure and alloyed metals. The authors find that two descriptors related to the metal emerge from the PCA, one that correlates with the d-band center and one that correlates with the standard reduction potential of the metal. Hence, the descriptors are separately capturing the ability of the metal to form covalent and ionic bonds, respectively. This application of PCA/R to determine data-driven descriptors in catalysis is novel (to the best of my knowledge) and here the authors demonstrate that it can be quite effective for reducing the DFT load in computational screening efforts. However, this is not the first study to use ML-based approaches to improve on

the d-band center model. In particular, the authors should cite recent work reported by Andersen et al. (ACS Catal.2019942752-2759), which is highly relevant to the work presented in this manuscript. Both studies use ML-based tools to improve upon the d-band center model when predicting adsorption trends across pure metals and alloys. The ML approach reported in the submitted manuscript is quite different than the one applied by Andersen et al. Some discussion comparing the advantages and disadvantages of the two approaches would be very informative for readers. Finally, there are some details regarding the methodology that I found difficult to follow, which should be clarified (details below). Overall, this paper should be of high interest to a general audience. For these reasons, I recommend that the manuscript be published after the completion of minor revisions and clarifications.

- In the main text the authors provide a concise summary of the steps taken in the PCA approach and then a more detailed explanation of the methodology is provided as supporting information. In the main text, the authors launch into a step-by-step summary of how the PCA was executed but provide readers with very little context for what the PCA is actually accomplishing. The manuscript would be improved if the authors first explained in the main text the central concepts that underlie PCA and why it is an appropriate tool for tackling the problem at hand. This would make the manuscript much more accessible to a wide audience, as many readers will not be familiar with PCA.
- There appears to be an inconsistency in Equation 4. The first term on the RHS has a prime on "t_{i1}" but then the prime is dropped in the following terms.
- In figure 3 the authors omit the data points for Cd and Zn because the d-band model breaks down for these elements. This should be stated more clearly in the figure caption, which currently only reads "additional plots and data are provided in Supplementary Figure 1." I suggest that the authors move the inset from Supp. Figure 1 that contains Cd and Zn data points to Figure 3 of the main text, so that readers can clearly see that the d-band model breaks down for Cd and Zn. In fact, this is a good argument for why the data-driven descriptors are an improvement over the d-band model, as they presumably capture data points that depart from the d-band trend.
- I had trouble following how the authors determined the contents of the training sets for alloy surfaces, which were used to generate the models reported in Figure 5. This should be clarified.

Answer to Reviewer #1:

We would like to thank the Reviewer for his/her positive assessment of our work. In the following his/her comments are in black, our answers in blue, and the actions taken in italics.

1. The chemistry involved in upgrading biomass-derived molecules is complex. The approach is not demonstrated to be useful for catalyst design rather than showing the prediction of binding energies of big fragments using simple groups, e.g., *C, *OH, and *OCH₂CH₂O*.

The workflow to achieve catalytic design from first principles has four steps: (i) evaluation of the thermodynamics of intermediates in all possible competing routes; (ii) obtention of the kinetic parameters from transition states; (iii) interpretation of kinetic data under operation conditions *via* a microkinetic model; (iv) development of selectivity and activity volcanoes to find the optimal catalyst candidates. In this hierarchical methodology, the first step already requires more than 70 DFT energy evaluations to evaluate the reactivity of a C₂ molecule on each alloy [*Nat. Commun.* **2018**, 9, 526]. Our approximation reduces the number of energy evaluations by 20 times while maintaining the robustness of the prediction. Thus, while our work does not complete the full value chain in catalysis design, it does provide the first step towards the robust use of statistical learning in heterogeneous catalysis. *We have highlighted the importance of reducing the number of energy evaluations in the rational design of heterogeneous catalysts: Page 2, Lines 34-38; Page 3, Lines 76-78.*

2. PCA analysis is not neat for dimensionality reduction. The two principle components that maximize the variance of adsorption energies are correlated to the *d*-band center and reduction potentials, respectively. The result is very interesting. However, relating the two descriptors of surfaces to two characteristics of metals is arbitrary and not necessary captures underlying physics.

Principal Component Analysis is a simple approach to reduce dimensionality and facilitate data interpretability [Hastie *et al.* *The Elements of Statistical Learning*, 2nd Edition. Springer. Pages 408-409; 536-537]. The two set of descriptors for metals and molecules, t_{jk} and w_{kj} in Equation 3:

$$\hat{E}_{ij} = t_{i1}w_{1j} + t_{i2}w_{2j} + \dots + t_{i k_{\max}} w_{k_{\max} j} + \mu_j \quad (3)$$

are obtained by projecting the dataset into the directions of largest variance, thus representing the main causes of the variability in metal-adsorbate bond energies. Hence, as the Reviewer indicates, t_{jk} and w_{kj} might not necessarily map to a unique characteristic. However, as the bond of an organic moiety to a metal can be understood as formed by covalent, ionic, and dispersion contributions and thus our idea was to map the descriptors to these contributions. Comparing the t_{j1} with the well-known *d*-band center gave us the idea that this was the descriptor for covalency, and

therefore we proceeded to look for the missing second term, the ionic contribution as the dispersion for such small molecules is not so relevant (and likely mostly follows the covalent term). *We added a sentence to clarify that we interpreted the descriptors: Page 6, Line 144; Page 8, Line 155.* During our analysis, we found that both covalency and ionicity were rather independent, *as stated now on Page 8, Lines 156-158.*

3. On page 8, it states that the conjugation and conformational effects are already wrapped up in the two descriptors. Please explain clearly.

In the datasets the ground state conformation for each adsorbate was always considered. For instance, CHCH₂ can adsorb as monodentate, tridentate, or intermediate structures as shown below:

The most stable conformation depends mainly on the covalent contribution of the metal. Therefore, the descriptors obtained upon applying PCA, t_{ik} and w_{kj} , already contain the variability of both conjugation and conformation. *We have highlighted the importance of choosing the ground state configuration of the adsorbates to obtain descriptors accounting for conjugation and conformation: Page 8, Lines 159-165.*

4. Three components are chosen later for capturing a broad range of systems. But on page 9, it says the estimation can be done with two principle components obtained from the formation energies of three surface species. It is not clear what they mean. To me, the number of descriptors for surface has to be same length as the number of surface species required for estimation.

In both Principal Component Analysis and Regression, PCA/PCR, we have used two principal components ($t_{1j}w_{1j}$ and $t_{2j}w_{2j}$). In the particular case of PCR, both pairs of descriptors were estimated from the formation energy of three predictors (O*, *OH, and *CCHOH). Hastie *et al.* in *The Elements of Statistical Learning*, 2nd Edition Springer pages 79-80, shows that the number of predictors, p , can be greater than the number of principal components, $p \geq k_{\max}$ [see also, *J. Phys. Chem. C* **2018**, 122, 28142-28150]. *We have added a note in the Manuscript to clarify that $p \geq k_{\max}$: Page 9, Lines 175-176. We have also summarized that we obtained two pairs of descriptors from three predictors: Page 10, Lines 191-195.*

Answer to Reviewer #2:

We would like to thank the Reviewer for his/her examination of our work. In the following, the only minor remark by the Reviewer is addressed. His/her comment is in black, our answer in blue, and the action taken in italics.

1. I think a better description on the alloys and single atoms would help understand what they did -the presentation is vague to the point where we do not know what systems etc-.

The single-atom alloys (SAA) were generated by replacing one atom on top of the surface by the guest element, **Figure 5 a** (inset). Conversely, for the near-surface alloys (NSA) either the overlayer (OL) or the sub-surface (SS) layer was substituted by the guest element, **Figure 5 b-c**. The guest elements were the 12 pure metals used along the manuscript, plus Fe, Co, and Re, for a total of $12 \times (15 - 1) = 168$ alloys of each kind. For each alloy, the DFT dataset contains the energy of the clean surface plus the three predictors (*O, *OH, and *CCHOH). There were 61 alloys whose structures did not converge. They were removed from the pool leaving a total of 165 SAA and 278 NSA. Then, using PCR and the 12 pure metals as training set, we predicted the thermochemistry of the remaining 68 adsorbates on each alloy. Around 1% of these structures were explicitly calculated by DFT and used as validation set. The benchmark is plotted in **Figure 5 b-d**. Following the Reviewer's suggestion, *we have improved the description of the single-atom and near-surface alloys by: (i) explaining how they were generated and their exact number: Page 12, Lines 239-246; (ii) listing the non-converged structures in the Supplementary Methods; (iii) recalling the training set composed by the pure metals, Page 13, Lines 249-250; and (iv) describing the validation set for SAA and NSA and how it was generated: Page 13, Lines 253-255.*

Figure 5 | (...) **b** Single-atom (SAA), **c** sub-surface (NSA-SS), and **d** overlayer (NSA-OL) alloys. Taking the DFT thermochemistry of 71 adsorbates on 12 pure metals and three predictors for each alloy (*O, *OH, and *CCHOH), the adsorption energies for the remaining 68 adsorbates was predicted. Then, 1% of these species was randomly selected as validation set, calculated by DFT, and benchmarked on panels **b-d**.

Answer to Reviewer #3:

We thank the Reviewer for his/her positive view on our manuscript. In the following his/her comments are in black, our answers in blue, and the actions taken in italics.

1. This is not the first study to use ML-based approaches to improve on the *d*-band center model. In particular, the authors should cite recent work reported by *Andersen et al.* (*ACS Catal.* **2019**, *9*(4), 2752-2759), which is highly relevant to the work presented in this manuscript. Both studies use ML-based tools to improve upon the *d*-band center model when predicting adsorption trends across pure metals and alloys. The ML approach reported in the submitted manuscript is quite different than the one applied by *Andersen et al.* Some discussion comparing the advantages and disadvantages of the two approaches would be very informative for readers.

The publication of the suggested article was simultaneous with the submission of our manuscript. *Andersen et al.* [*ACS Catal.* **2019**, *9*, 2752-2759] applied a compressed-sensing algorithm (SISSO) that predicts adsorption energies by taking features from the adsorbate, the metal, and the adsorption site. As a comparative advantage, their method requires only one DFT evaluation for each new surface while ours requires four. However, the SISSO approach is purely predictive while our approach tends to be in line with Explanatory Machine Learning. Following the Reviewer's suggestion, *we cite and compare the methodology of Andersen et al. as Reference #45: Page 4, Lines 84-88.*

2. In the main text the authors provide a concise summary of the steps taken in the PCA approach and then a more detailed explanation of the methodology is provided as supporting information. In the main text, the authors launch into a step-by-step summary of how the PCA was executed but provide readers with very little context for what the PCA is actually accomplishing. The manuscript would be improved if the authors first explained in the main text the central concepts that underlie PCA and why it is an appropriate tool for tackling the problem at hand. This would make the manuscript much more accessible to a wide audience, as many readers will not be familiar with PCA.

We are aware that many groups are trying to introduce statistical learning methods and this addition will indeed help in the reproducibility of our work. PCA is a statistical technique that reduces the dimensionality of a data matrix by projecting it along the directions of greatest variability, thus reducing its noise and aiding to its interpretability. It can only be applied on complete data matrices, i.e., those that do not have any missing points, like the thermochemistry matrix **E**. Consequently, following the Reviewer's advice, *we have added a short conceptual framework of what is the object and scope of PCA, Page 5, Lines 114-118. We have also expanded the description about how we used it, Lines 118-127.*

3. There appears to be an inconsistency in Equation 4. The first term on the RHS has a prime on “ t_{i1} ” but then the prime is dropped in the following terms.

We thank the Reviewer for carefully checking our Equations. Indeed, Equation 4 is based on matrix \mathbf{T} , which contains the thermochemistry of the three predictors on the training set. *We have amended Equation 4 and double-checked the remaining ones:*

$$E''_{ij} = t'_{i1}w_{1j,\text{val}} + t'_{i2}w_{2j,\text{val}} + \dots + t'_{ik_{\text{max}}}w_{k_{\text{max}}j,\text{val}} + \mu_{j,\text{val}}$$

4. In figure 3 the authors omit the data points for Cd and Zn because the d-band model breaks down for these elements. This should be stated more clearly in the figure caption, which currently only reads “additional plots and data are provided in Supplementary Figure 1.” I suggest that the authors move the inset from Supp. Figure 1 that contains Cd and Zn data points to Figure 3 of the main text, so that readers can clearly see that the d-band model breaks down for Cd and Zn. In fact, this is a good argument for why the data-driven descriptors are an improvement over the d-band model, as they presumably capture data points that depart from the d-band trend.

As the Reviewer points out, the *d*-band model alone cannot describe some systems that are relevant for catalysis. One of the groups that do not follow the typical *d*-band trends are adsorbates with almost-filled valence shells on metals with almost-filled *d*-bands [*J. Chem. Phys.* **2010**, 132, 221101]. Another group are Cd, Zn, and post-transition metals that can be components in high-entropy alloys suited for electrocatalysis [*Joule*, **2019**, 3, 834-845]. In the light of Reviewer’s suggestion, *we now reflect this limitation on Page 8, Lines 147-152. We have also moved the inset from Supplementary Figure 1 into Figure 3 and edited the caption accordingly.*

Figure 3 a | First descriptor for the metals, t_1 , plotted against the *d*-band center. The inset shows that *d*-band center controls the adsorption energy on late transition metals (groups 8-11) [*Nature* **1995**, 376, 238-240; *Phys. Rev. Lett.* **1996**, 76, 2141] but it cannot describe the behavior of Zn and Cd.

5. I had trouble following how the authors determined the contents of the training sets for alloy surfaces, which were used to generate the models reported in Figure 5. This should be clarified.

We thank the Reviewer for the comment, as the description of the alloys was not clear. The single-atom alloys (SAA) were generated by replacing one atom on top of the surface by the guest element, **Figure 5 a** (inset). Conversely, for the near-surface alloys (NSA) either the overlayer (OL) or the sub-surface (SS) layer was substituted by the guest element, **Figure 5 b-c**. The guest elements were the 12 pure metals used along the manuscript, plus Fe, Co, and Re, for a total of $12 \times (15 - 1) = 168$ alloys of each kind. For each alloy, the DFT dataset contains the energy of the clean surface plus the three predictors (*O, *OH, and *CCHOH). There were 61 alloys whose structures did not converge. They were removed from the pool leaving a total of 165 SAA and 278 NSA. Then, using PCR and the 12 pure metals as training set, we predicted the thermochemistry of the remaining 68 adsorbates on each alloy. Around 1% of these structures were explicitly calculated by DFT and used as validation set. The benchmark is plotted in **Figure 5 b-d**. Following the Reviewer's suggestion, we have improved the description of the single-atom and near-surface alloys by: (i) explaining how they were generated and their exact number: Page 12, Lines 239-246; (ii) listing the non-converged structures in the Supplementary Methods; and (iii) describing the validation set for SAA and NSA and how it was generated: Page 13, Lines 253-255.

Figure 5 | (...) **b** Single-atom (SAA), **c** sub-surface (NSA-SS), and **d** overlayer (NSA-OL) alloys. Taking the DFT thermochemistry of 71 adsorbates on 12 pure metals and three predictors for each alloy (*O, *OH, and *CCHOH), the adsorption energies for the remaining 68 adsorbates was predicted. Then, 1% of these species was randomly selected as validation set, calculated by DFT, and benchmarked on panels **b-d**.

REVIEWERS' COMMENTS:

Reviewer #1 (Remarks to the Author):

The two principal components identified from PCA are related to the d-band center and reduction potential of metals. The d-band center is clearly defined value. What is reduction potential of metals used? Can we just use the electronegativity of metals, e.g. Pauling or Mulliken? All other questions and concerns are clearly addressed.

Reviewer #3 (Remarks to the Author):

The authors have adequately responded to all of my previous comments and have improved the manuscript accordingly. This work is now suitable for publication.

We would like to thank the Reviewer for his/her last comment on our work. In the following, the only minor remark by the Reviewer is addressed. His/her comment is in black, our answer in blue, and the action taken in italics.

The two principal components identified from PCA are related to the d-band center and reduction potential of metals. The d-band center is clearly defined value. What is reduction potential of metals used? Can we just use the electronegativity of metals, e.g. Pauling or Mulliken? All other questions and concerns are clearly addressed.

Based on the good correlation between t_{2z} and the reduction potential, E_h , we interpreted the second principal component, $t_{2z} w_{2j}$, as a summary of the ionic contributions to the adsorption energy. However, as the Reviewer suggest, other measures of ionic contributions can be used as well.

The Pauling electronegativity of a given element, χ_A in $\text{eV}^{-0.5}$, is defined from valence bond theory as the additional stabilization of a heteronuclear bond (A–B) due to the ionic contribution, Equation R1. The left side of Equation R1 has one degree of freedom, thus fluorine is arbitrarily assigned as $\chi_F = 4 \text{ eV}^{-0.5}$ to fix the scale.

$$|\chi_A - \chi_B| = \sqrt{E_{A-B} - \frac{1}{2}E_{A-A} - \frac{1}{2}E_{B-B}} \quad (\text{R1})$$

Mulliken electronegativity, χ'_A in eV, is defined as the average of electron affinity and ionization potential. The later quantities are defined for isolated neutral atoms in gas phase, instead of bulk metals.

$$\chi'_A = \frac{1}{2}E_{\text{ea},A} + \frac{1}{2}E_{\text{ip},A} \quad (\text{R2})$$

We have plotted the second metal descriptor, t_{2z} , as a function of Pauling and Mulliken electronegativities in Supplementary Figure 5. Both quantities support the conclusion that the second principal component, $t_{2z} w_{2j}$, summarizes the ionic contributions. However, both correlations are poorer than when the reduction potential is used, because the units of Pauling electronegativity are not consistent with t_{2z} , and Mulliken electronegativity is defined for metals in gas phase, instead of bulk.

Supplementary Figure 5: Second descriptor for the metals, t_{2z} , plotted against **a** Pauling and **b** Mulliken electronegativity.